# Recovering Power Grids Using Strategies Based on Network Metrics and Greedy Algorithms

**DOI:** 10.3390/e25101455

**Published:** 2023-10-17

**Authors:** Fenghua Wang, Hale Cetinay, Zhidong He, Le Liu, Piet Van Mieghem, Robert E. Kooij

**Affiliations:** 1Faculty of Electrical Engineering, Mathematics and Computer Science, Delft University of Technology, 2628 CD Delft, The Netherlands; l.liu-7@tudelft.nl (L.L.); p.f.a.vanmieghem@tudelft.nl (P.V.M.); r.e.kooij@tudelft.nl (R.E.K.); 2Asset Management, System Insights and Advanced Analytics, Stedin, 3011 TA Rotterdam, The Netherlands; halecetinay@outlook.com; 3DS Information Technology Co., Ltd., Shanghai 200032, China; zhidonghe@outlook.com; 4Unit ICT, Strategy and Policy, Netherlands Organisation for Applied Scientific Research (TNO), 2595 DA Den Haag, The Netherlands

**Keywords:** power grids, network resilience, network recoverability

## Abstract

For this study, we investigated efficient strategies for the recovery of individual links in power grids governed by the direct current (DC) power flow model, under random link failures. Our primary objective was to explore the efficacy of recovering failed links based solely on topological network metrics. In total, we considered 13 recovery strategies, which encompassed 2 strategies based on link centrality values (link betweenness and link flow betweenness), 8 strategies based on the products of node centrality values at link endpoints (degree, eigenvector, weighted eigenvector, closeness, electrical closeness, weighted electrical closeness, zeta vector, and weighted zeta vector), and 2 heuristic strategies (greedy recovery and two-step greedy recovery), in addition to the random recovery strategy. To evaluate the performance of these proposed strategies, we conducted simulations on three distinct power systems: the IEEE 30, IEEE 39, and IEEE 118 systems. Our findings revealed several key insights: Firstly, there were notable variations in the performance of the recovery strategies based on topological network metrics across different power systems. Secondly, all such strategies exhibited inferior performance when compared to the heuristic recovery strategies. Thirdly, the two-step greedy recovery strategy consistently outperformed the others, with the greedy recovery strategy ranking second. Based on our results, we conclude that relying solely on a single metric for the development of a recovery strategy is insufficient when restoring power grids following link failures. By comparison, recovery strategies employing greedy algorithms prove to be more effective choices.

## 1. Introduction

The power grid system is vulnerable to disruptions caused by manufacturing defects, natural disasters, and human actions, such as terrorist attacks [1,2,3]. Even minor initial disturbances can lead to severe consequences, including economic and social instability [4,5,6]. To reduce the costs associated with the disruptions, one research direction is to enhance the robustness of the power grid to withstand perturbations. The other direction is to propose efficient restoration strategies.

The power grid can be represented as a complex network, where nodes represent generators and loads, and where links represent transmission lines and transformers. Analyzing the power grid’s robustness from a network perspective has attracted significant attention. Network robustness is typically evaluated by assessing changes in network performance due to fluctuations such as node removals or link removals [7]. Identifying critical nodes and links within the power grid can inform strategies for enhancing robustness by protecting the vital components [8,9,10]. Furthermore, robust failure responses, like partitioning grids into islands, can mitigate the effects of components disruptions [11,12]

Numerous studies have focused on strategies for power grid recovery [13], from black start [14]—like an optimal generator start-up strategy by solving a mixed integer linear programming (MILP) problem [15]—to a method that partitions the network to restore parts of the power grid separately and then interconnects them afterward [16]. Additionally, research has been conducted on recovering power grids after partial component failures. Machine learning methods, specifically reinforcement learning, have been developed for restoring networks after node failures [17] and link failures [18]. The machine learning methods demonstrated better performance for node recovery strategies based on node degrees or node loads and link recovery strategies based on link betweenness. Li et al. [19] developed the Q-learning method, to find an optimal method to recover power grids with link failures. From a network perspective, Wu et al. [20] developed an effective tool for the sequential recovery graph, to recover nodes in power grids, which performed better than recovery strategies based on node degree and node loads. Forming microgrids can improve the resilience of the system after blackouts. Igder et al. [21] applied deep reinforcement learning, to establish microgrids from black start after blackouts, so as to restore service in distribution networks. Yeh et al. [22] developed an enhanced genetic algorithm, to minimize costs while optimizing dispatch in stand-alone microgrid systems.

Despite extensive research on power grid recovery, there is still a lack of investigation into the effectiveness of recovery strategies that rely solely on different network metrics following transmission line failures. From this study, our main contributions are as follows:We examined recovery strategies based on various network metrics, including degree, betweenness, flow betweenness, eigenvector centrality, weighted eigenvector centrality, closeness, electrical closeness, electrical weighted closeness, zeta vector centrality, and weighted zeta vector centrality. Additionally, we compared these strategies to the random recovery, greedy, and two-step greedy strategies.To assess the effectiveness of recovery methods, we utilized the general recoverability framework proposed by He et al. [23], to measure power grid recoverability in the context of random link removals, where recoverability signifies a network’s ability to return to a predefined desired performance level.Our study did not consider cascading failures after the recovery or removal of a single transmission line. Instead, we used the direct current (DC) power flow model to maximize power flow satisfaction for loads after a link removal or addition.

The paper is structured as follows: Section 2 provides an overview of network robustness, laying the foundation for the subsequent analyses. Section 3 details the modeling of power grids, including how to transfer a power grid into a network and optimize the DC power flow model. Section 4 presents the attack and recovery processes, with a focus on the strategies employed. The results and conclusion are presented in Section 5 and Section 6, respectively.

## 2. Preliminary for Network Robustness

### 2.1. *R*-Value and Challenges

In a framework for computing network robustness [7], network robustness is interpreted as a measure of the network’s response to perturbations, such as failures or attacks. The robustness value *R* was proposed to measure the performance of a network at a certain time, which is related to the function of the network, i.e., the type of service the network is supposed to support, such as road transport, neuron transport or the spreading of news on a social network. The *R*-value is normalized in the range [0,1]. Here, R=0 corresponds to a network completely lacking robustness, while R=1 corresponds to an optimally robust network.

We assume that perturbations are imposed on a network through a number of elementary changes [7,23,24]. We denote the total number of the changes by *K*. An elementary change in a network is defined as an event that changes the topology of a network, such as a link addition, a link removal or a change in the link weight. An elementary change in the network may result in a change of the *R*-value. For this paper, we considered elementary changes as the random removal of a link during the attack phase and the recovery of a link previously removed in the attack phase. Additionally, we denoted the number of challenges in the attack process as $K_{a}$ and in the recovery process as $K_{r}$.

### 2.2. Recoverability Indicator of a Recovery Strategy

The performance of a power grid at any time can be captured by a specific *R*-value, which possibly changes when an elementary change occurs. The *R*-value at challenge *k* is denoted as R[k]. The areas under the *R*-value line during the processes can reflect the effectiveness of the attack strategy or the recovery strategy [25], which is shown in Figure 1. We define the attack strength $S_{a}$ and the recovery strength $S_{r}$, which satisfy the following relations:(1)$$S_{a}=\sum_{k=0}^{K_{a}}R\left[k\right]$$
and
(2)$$S_{r}=\sum_{k=K_{a}}^{K_{a}+K_{r}}R\left[k\right],$$
where $K_{a}$ is the total number of challenges in the attack process and $K_{r}$ is the total number of challenges in the recovery process. Furthermore, we define the recoverability energy ratio η, as follows:(3)$$\eta =\frac{S_{r}}{S_{a}}.$$

The recoverability energy ratio η presents the efficiency of the recovery strategy, with respect to the attack strategy. The larger the recoverability energy ratio η is, the more efficient the recovery strategy is.

## 3. Modeling Power Grids

### 3.1. Network Model of Power Grids

A power grid can be represented by a graph G, where the set of *N* nodes is denoted as N and the set of *L* links is denoted as L. For an unweighted and undirected graph, the symmetric adjacency matrix *A* has elements $a_{ij}=1$ if the nodes *i* and *j* are connected by a link $l_{ij}\in L$, otherwise, $a_{ij}=0$. Alternatively, we can also model a power grid as a weighted graph. The link weight is related to the impedance of transmission line $l_{ij}$, which will be denoted as $y_{ij}$. The weighted symmetric adjacency matrix $\tilde{A}$ has elements ${\tilde{a}}_{ij}=\frac{1}{y_{ij}}$ if a link $l_{ij}\in L$ exists, otherwise, ${\tilde{a}}_{ij}=0$ [26].

For this study, we chose three power grids [27]: IEEE 30, IEEE 39, and IEEE 118. We present the number of nodes *N*, the number of links *L*, and the average degree $d_{av}$ of the unweighted power grids in Table 1.

### 3.2. Performance of Power Grids

As proposed by Cetinay et al. [28], we employ an *R*-value in a power grid: the ratio of the total satisfied demand to the total initial demand, called the yields. At challenge *k*, we denote the amount of satisfied demand at bus *i* by $L_{i}\left[k\right]$ and the amount of supply at bus *i* by $G_{i}\left[k\right]$. Therefore, the injected power $P_{i}$ at node *i* satisfies $P_{i}\left[k\right]=G_{i}\left[k\right]-L_{i}\left[k\right]$. As the total demand matches the supply in a power grid, it holds that $\sum_{i=1}^{N}L_{i}\left[k\right]=\sum_{i=1}^{N}G_{i}\left[k\right]$. At challenge k=0, the power grid has not been attacked and the initial demand of bus *i* is $L_{i}\left[0\right]$. Then, the yields—an *R*-value at a challenge *k*—can be calculated by
(4)$$R\left[k\right]=\frac{\sum_{i=1}^{N}L_{i}\left[k\right]}{\sum_{i=1}^{N}L_{i}\left[0\right]}.$$

### 3.3. Optimizing the DC Power Flow Model

Compared to cascading failure models in power grids, we propose a power flow redistribution mechanism to achieve a steady state by optimizing the total satisfied demand value. The mechanism adjusts the supply and demand of buses, using a linear programming method to ensure that line flows are below the line capacity. Specifically, the objective of the proposed model is to optimize the total satisfied demand while satisfying several constraints, including: (1) the power grid satisfies Kirchhoff’s Law and Ohm’s Law; (2) the total supply matches the total demand at each challenge; (3) the supply and demand of a bus are not beyond its initial values; (4) the absolute value of a line flow is not beyond the line capacity. The DC power flow model [28] at challenge *k* obeys the optimization situation:(5)$$\begin{array}{c} \mathrm{minimize}\;\;\;\;\;\;-\sum_{i=1}^{N}G_{i}\left[k\right]\;\\ \mathrm{subject}\;\mathrm{to}\;\;\;\;\;\;G\left[k\right]-L\left[k\right]=\tilde{Q}\left[k\right]\Theta \left[k\right],\;\\ \;\;\;\;\;\;\;\;\;\;\;\;\;\;\;\;\;\;\;\;F\left[k\right]={\tilde{B}}^{T}\left[k\right]\Theta \left[k\right],\;\\ \;\;\;\;\;\;\;\;\;\;\;\;\;\;\;\;\;\;\;\;\sum_{i}^{N}G_{i}\left[k\right]-\sum_{i}^{N}L_{i}\left[k\right]=0,\;\\ \;\;\;\;\;\;\;\;\;\;\;\;\;\;\;\;\;\;\;\;0\leq G[k]\leq G[0],\;\\ \;\;\;\;\;\;\;\;\;\;\;\;\;\;\;\;\;\;\;\;0\leq L[k]\leq L[0],\;\\ \;\;\;\;\;\;\;\;\;\;\;\;\;\;\;\;\;\;\;\;-C\leq F[k]\leq C. \end{array}$$
The weighted *Laplacian* matrix is $\tilde{Q}\left[k\right]=\tilde{\Delta }\left[k\right]-\tilde{A}\left[k\right]$, where $\tilde{\Delta }\left[k\right]$ is the weighted degree matrix and $\tilde{A}\left[k\right]$ is the weighted adjacency matrix; Θ is the N×1 vector with elements $\theta_{i}$, which presents the phase angle of bus *i*; $\tilde{B}\left[k\right]$ is the N×L weighted incidence matrix with elements
(6)$${\tilde{b}}_{il}=\left\{\begin{array}{cc} {\tilde{a}}_{ij} & \;\mathrm{if}\;\mathrm{link}\;e_{l}=i\to j,\\ -{\tilde{a}}_{ij} & \;\mathrm{if}\;\mathrm{link}\;e_{l}=i\leftarrow j,\\ 0 & \;\mathrm{otherwise}, \end{array}\right.$$
and $\tilde{Q}\left[k\right]=\tilde{B}\left[k\right]\tilde{B}{\left[k\right]}^{T}$.

The supply and demand vectors G[k] and L[k] include the supply $G_{i}\left[k\right]$ and demand $L_{i}\left[k\right]$ of each bus *i*, respectively. The initial supply and demand values of all buses are stored in the vectors G[0] and L[0] at challenge k=0. The active power flow vector F[k] has *L* elements, each representing the power flow $f_{ij}\left[k\right]$ of a transmission line $l_{ij}$ connecting buses *i* and *j*. The capacity vector C is a vector with *L* elements and each element $c_{ij}$ representing the capacity of each transmission line $l_{ij}$.

For the power grid, the line capacity $c_{ij}$ is defined as $\alpha f_{ij}\left[0\right]$, where flow vector F[0] is the active power flow of each line at the initial stage (at challenge k=0) and α is the tolerance level [29].

## 4. The Attack and Recovery Process

### 4.1. The Attack Process

In the attack process, we select links uniformly at random, to be removed iteratively until the *R* value of the power grid falls below a predetermined threshold. The flowchart of the attack process is shown in Figure 2a:

### 4.2. The Recovery Process

In the recovery process, links previously removed from the power grid are gradually added back individually, with the order of additions determined by a chosen recovery strategy. The flowchart of the recovery process is presented in Figure 2b. We explore the efficiency of 13 different recovery strategies introduced in the following section. It is important to note that strategies based on topological network metrics are calculated based on the original network configuration before the attack process. The flow chart of recovery strategies based on network metrics is presented in Figure 3.

#### 4.2.1. Random Recovery Strategy (Rand)

A link is randomly selected, to add to the power grid one by one from the set of links that were removed during the attack process.

#### 4.2.2. Greedy and Two-Step Greedy Recovery Strategies (Greedy and TwoGreedy)

The greedy recovery strategy is to optimize the performance of the power grid at each stage of the recovery process. The strategy selects the link that yields the largest *R*-value of the power grid in each step and adds the link to the grid. The greedy recovery strategy flow chart is depicted in Figure 4a.

In contrast to the greedy recovery strategy, which focuses on identifying a single link that significantly improves the *R*-value in one step, an improved approach involves selecting *n* links that collectively maximize the summation of the *R*-value in *n* steps, which can be achieved by exhaustively enumerating all permutations of *n* links from the set of potential links to be added. By considering multiple steps, more information is incorporated, resulting in potentially better solutions for the *n*-step greedy recovery strategy. However, if the value of *n* becomes excessively large, the computational cost becomes prohibitive. Therefore, in this study, we chose to investigate the performance of the two-step greedy recovery strategy. Specifically, the two-step greedy strategy aims to determine two links to be added sequentially, optimizing the sum of the *R*-values achieved in the two steps. We present the flowchart of the two-step greedy strategy in Figure 4b.

#### 4.2.3. Degree Recovery Strategy (Degree)

The degree $d_{i}$ is the number of neighboring nodes of node *i*, which can be used to measure the node’s importance. By using the adjacency matrix *A*, the calculation of the degree is $d_{i}=\sum_{j=1}^{N}a_{ij}$. To evaluate the significance of a link, we can use the product $d_{ij}$ of the degrees of the nodes connected by the link $l_{ij}$ [30], denoted as $d_{ij}=d_{i}d_{j}$.

The degree recovery strategy involves a sequence of link additions, based on the descending order of the product of the degrees of the end points of each removed link. Consequently, the first link to be added possesses the largest product of the degrees of its end points in the removed link set.

#### 4.2.4. Betweenness and Flow Betweenness Recovery Strategies (Bet and FlowBet)

The betweenness of a link is widely used to measure the importance of a link $l_{ij}$ in the network. It is defined as the ratio of the shortest paths through the link $l_{ij}$ to the number of all shortest paths in the network [31]. If we denote the number of shortest paths from node *s* to node *t* as $P_{s\to t}$ and the number of the shortest paths from node *s* to node *t* through the link $l_{ij}$ as $P_{s\to t}\left(l_{ij}\right)$, the betweenness $b_{ij}$ of link $l_{ij}$ can be calculated by
(7)$$b_{ij}=\sum_{s,t\in N}\frac{P_{s\to t}\left(l_{ij}\right)}{P_{s\to t}}.$$

The shortest paths are the most efficient ways for a flow in a network to travel from one node to another node if the traveling cost of every link is the same and there are no other limitations in the network, like link capacity. However, in a power grid, the distribution of the power flow is determined by the Kirchhoff and Ohm laws, not by the shortest paths. Therefore, Newman [32] proposed flow betweenness to measure the link importance in a power grid. The flow betweenness ${\bar{b}}_{ij}$ of link $l_{ij}$ is defined as the sum of the power flow through link $l_{ij}$ for any node pair *s* and *t* if one unit of power is injected to node *s* and one unit of power is extracted from node *t*. The flow betweenness ${\bar{b}}_{ij}$ of link $l_{ij}$ can be calculated by
(8)$${\bar{b}}_{ij}=\sum_{s,t\in N}\left|f_{s\to t}\left(l_{ij}\right)\right|,$$
where $|f_{s\to t}\left(l_{ij}\right)|$ is the magnitude of flow through the link $l_{ij}$ according to DC flow equations and Ohm’s law when we inject one unit of active power to node *s* and extract one unit of active power from node *t*.

The sequence of added links in the betweenness or flow betweenness recovery strategies assumes a descending order of betweenness or flow betweenness of links.

#### 4.2.5. Eigenvector and Weighted Eigenvector Recovery Strategies (Eigen and WeiEigen)

The eigenvector centrality measures the importance of a node by not only taking into account the number of its connected nodes but also considering the importance of its connected nodes [30,33]. The eigenvector centrality $x_{i}$ of node *i* is the *i*-th element of the eigenvector with the largest eigenvalue of the adjacency matrix *A*. Compared to the eigenvector centrality, the weighted eigenvector centrality ${\bar{x}}_{i}$ of node *i* is the *i*-th element of the eigenvector with the largest eigenvalue of the weighted adjacency matrix $\tilde{A}$. The products $e_{ij}$ and ${\bar{e}}_{ij}$ of the eigenvector centrality and the weighted eigenvector centrality of link $l_{ij}$’s end points are calculated by $e_{ij}=x_{i}x_{j},{\bar{e}}_{ij}={\bar{x}}_{i}{\bar{x}}_{j}.$

The sequences of links are added in the descending order of the products $e_{ij}$ or ${\bar{e}}_{ij}$ of the links in the eigenvector or the weighted eigenvector recovery strategies.

#### 4.2.6. Closeness, Electrical Closeness, and Electrical Weighted Closeness Recovery Strategies (Close, EleClose, and EleWeiClose)

The closeness centrality of a node relates to the node’s distance from all other nodes [34]. If the distance of the shortest path between node *i* and node *j* is denoted by $H_{ij}$, then the closeness ${\bar{c}}_{i}$ of node *i* is defined as
(9)$${\bar{c}}_{i}=\frac{1}{\sum_{j=1}^{N}H_{ij}}.$$

We calculate the product ${\bar{c}}_{ij}$ of the closeness of link $l_{ij}$’s end points, ${\bar{c}}_{ij}={\bar{c}}_{i}{\bar{c}}_{j}$, which is used in the closeness recovery strategy, to measure the importance of a link.

As the flow in a power grid follows the Kirchhoff law—not only the shortest path—using so-called effective resistance is a more appropriate way to quantify the distance between a pair of nodes in a power grid [35]. The effective resistance between node *i* and node *j* is denoted as $\Omega_{ij}$, which can be calculated by using the pseudo-inverse matrix $Q^{†}$ of the Laplacian matrix *Q* of the adjacency matrix *A*: $\Omega_{ij}={\left(Q^{†}\right)}_{ii}+{\left(Q^{†}\right)}_{jj}-2{\left(Q^{†}\right)}_{ij}$ [36]. By analogy with closeness, the electrical closeness ${\tilde{c}}_{i}$ of node *i* is given by
(10)$${\tilde{c}}_{i}=\frac{1}{\sum_{j=1}^{N}\Omega_{ij}}.$$

We use the product ${\tilde{c}}_{ij}$ of the electrical closeness values of the end points of link $l_{ij}$ in the electrical closeness recovery strategy, where ${\tilde{c}}_{ij}={\tilde{c}}_{i}{\tilde{c}}_{j}$.

Compared to the electrical closeness of a link, the difference in calculating the electrical weighted closeness of a link is that we use the Laplacian matrix $\tilde{Q}$ of the weighted adjacency matrix $\tilde{A}$. Analogously, we firstly obtain the effective resistance $\tilde{\Omega }$ using ${\tilde{\Omega }}_{ij}={\left({\tilde{Q}}^{†}\right)}_{ii}+{\left({\tilde{Q}}^{†}\right)}_{jj}-2{\left({\tilde{Q}}^{†}\right)}_{ij}$, where ${\tilde{Q}}^{†}$ is the pseudo-inverse matrix of the Laplacian matrix $\tilde{Q}$. Then, the electrical weighted closeness $w_{i}$ of node *i* is calculated by
(11)$$w_{i}=\frac{1}{\sum_{j=1}^{N}{\tilde{\Omega }}_{ij}}.$$

The product $w_{ij}$ of the electrical weighted closeness values of the end points of link $l_{ij}$, $w_{ij}=w_{i}w_{j}$, is the measurement of a link in the electrical weighted closeness strategy.

In the closeness, electrical closeness, and electrical weighted closeness recovery strategies, we prioritize the recovery of links based on the descending order of the products ${\bar{c}}_{ij}$, ${\tilde{c}}_{ij}$ or $w_{ij}$.

#### 4.2.7. Zeta Vector and Weighted Zeta Recovery Strategies (Zeta and WeiZeta)

The zeta vector ζ, inspired by the electrical flow in a resistant network, serves as a representation of nodal spread capacity. The minimization of the zeta vector leads to the identification of the best spreader [26]. Specifically, the zeta vector consists of the diagonal elements of the pseudo-inverse matrix $Q^{†}$, derived from the Laplacian matrix *Q*. Thus, $\zeta =({\left(Q^{†}\right)}_{11},{\left(Q^{†}\right)}_{22},\ldots ,{\left(Q^{†}\right)}_{ii})$, where ${\left(Q^{†}\right)}_{ii}$ represents the zeta vector value $\zeta_{i}$ associated with node *i*.

When employing the weighted adjacency matrix $\tilde{A}$, the weighted zeta vector $\tilde{\zeta }$ is the diagonal elements of the pseudo-inverse matrix of the weighted Laplacian matrix $\tilde{Q}$, denoted as $\tilde{\zeta }=\left({\left({\tilde{Q}}^{†}\right)}_{11},{\left({\tilde{Q}}^{†}\right)}_{22},\ldots ,{\left({\tilde{Q}}^{†}\right)}_{ii}\right)$. Here, ${\left({\tilde{Q}}^{†}\right)}_{ii}$ represents the weighted zeta vector value ${\tilde{\zeta }}_{i}$ associated with node *i*.

To assess the significance of a link based on the zeta vector values of its end points, we introduce the zeta vector metric $\zeta_{ij}=\zeta_{i}\zeta_{j}$ and the weighted zeta vector metric ${\tilde{\zeta }}_{ij}=\tilde{\zeta_{i}}\tilde{\zeta_{j}}$ for a given link $l_{ij}$. The metrics are calculated by multiplying the zeta vector values or the weighted zeta vector values of the link end points.

To restore the removed links in the zeta vector and weighted zeta vector recovery strategies, we adopt a ranking scheme based on the descending order of the links’ zeta vector metric values or weighted zeta vector metric values.

## 5. Results

To investigate the effectiveness of recovery strategies in power grids, we conducted simulations on three different power grids: the IEEE 30 bus system, the IEEE 39 bus system, and the IEEE 118 bus system. In each realization, we randomly removed links until the *R*-value of the system fell below a specified threshold. We then used various strategies to restore the system until all the removed links were added. Finally, we computed the recoverability energy ratio of each recovery strategy, given the same random removal process. To explore the impact of thresholds and tolerance levels, we selected two thresholds (0.8 and 0.5) and four tolerance levels (1, 2, 2.5, and 3) and performed 1000 realizations for a power grid with each setting. In Figure 5, we demonstrate how the *R*-value varied in a realization with different recovery strategies for the IEEE 39 bus system and the abbreviations of the strategies used in the following figures and tables in Table 2.

We present the recoverability energy ratios of various recovery strategies concerning two thresholds and a tolerance level equal to 1 for the considered power grids, as shown in Table 3, Table 4 and Table 5. The tables illustrate that the greedy two-step method exhibited the highest mean value of recoverability energy ratio and the lowest standard deviation value for all three power grids in both threshold cases. Additionally, the greedy method consistently maintained the second-best performance, in terms of the recoverability energy ratio mean value for all three power grids, albeit its performance was slightly inferior to that of the greedy two-step method. Notably, the difference in performance between the greedy two-step method and the greedy method was more pronounced when the threshold was equal to 0.5, compared to the case when the threshold was 0.8. This observation suggests that the greedy two-step method outperforms the greedy method when the removal link set is relatively large. Another noteworthy finding concerned the results of the random recovery method. The mean recoverability energy ratios of all the power grids with different thresholds of the random recovery method were less than 1, indicating that the cost of recovery outweighed the cost of attacks.

For the recovery strategies utilizing link metrics or the product of node centralities of link end points, we observed noteworthy variations in performance across different power grids. Specifically, the recovery method based on link betweenness exhibited diverse rankings of mean recoverability energy ratio when applied to the three power grids under consideration. In the IEEE 30 bus system, the mean recoverability energy ratio rank of the betweenness recovery method fell within the bottom two. Conversely, within the IEEE 39 bus system, the betweenness recovery method achieved an intermediate rank among all the methods based on the mean recoverability energy ratio. Remarkably, the recovery method based on betweenness centrality yielded the highest mean recoverability energy ratio among all the recovery strategies based on ink metrics or the product of the node centrality values of link end points in the IEEE 118 bus system.

In addition to the betweenness recovery strategy, the zeta vector recovery strategy exhibited a similar performance pattern. The strategy demonstrated effectiveness in the IEEE 30 bus system, with a threshold value of 0.5, as well as in the IEEE 39 bus system. However, the efficiency diminished when applied to the IEEE 118 bus system. The results indicate that the efficacy of recovery methods based on link metrics or the product of node centralities of link end points is contingent upon the underlying network topology and dynamics of the networks.

The observation can be attributed to the distinct allocation of generators in the IEEE 39 bus system, as illustrated in Figure 6. Specifically, the majority of the generators in the IEEE 39 bus system had only one neighbor, and the proportion of links attached to generators was 22.74%, which was significantly higher than the corresponding proportions in the other two systems. By contrast, the IEEE 30 bus system had only one generator with a neighbor, and the proportion of links connected to generators was 14.63%, while the IEEE 118 bus system had three generators with a neighbor, and the proportion of links connected to generators was 10.62%. Therefore, the location of generators in the IEEE 39 system made the links connected to generators more susceptible to attacks and less likely to be recovered than the other two systems. Based on calculating the average values of the link metrics used for the different recovery strategies of two kinds of links—links connecting generators and loads, and links connecting loads and loads in Appendix A Table A2—we found that in the IEEE 39 system, the link metrics of the links connecting generators and loads were larger than the link metrics of the links connecting loads and loads in the zeta recovery strategy and weighted zeta recovery strategy, indicating that recovering the links connecting generators and loads matters in the recovery process.

Compared to the IEEE 30 bus system and the IEEE 39 bus system, the recovery methods based on link metrics and the product of node centrality values of link end points in the IEEE 118 system demonstrated substantially better performance. Specifically, most recovery strategies based on link metrics and the product of node centrality values of link end points exhibited higher average recoverability energy ratios than the random recovery strategy for both thresholds.

Link capacity is a crucial factor in causing blackouts in power grids. Although our method did not employ the cascading failure model, we investigated the impact of link capacity on recovery strategies. To this end, we conducted simulations with different tolerance levels (α=1, α=2, α=2.5, α=3), and we analyzed the results, which are presented in Figure 7 for the IEEE 30 bus system, the IEEE 39 bus system, and the IEEE 118 bus system. The simulation results show that the mean values of the recoverability energy ratio with respect to different tolerance levels did not change monotonically for the same recovery method, which could inspire study of the optimal tolerance level. The ranking of recovery strategies based on network metrics may have varied slightly, but the two-step greedy recovery strategy consistently performed the best, with the greedy recovery strategy following closely behind. Moreover, the results indicate that the recovery strategy based on betweenness outperformed the recovery strategy based on flow betweenness, while the electrical weighted closeness recovery strategy was the best among the electrical closeness recovery strategy, closeness recovery strategy, and electrical weighted closeness recovery strategy.

## 6. Conclusions and Discussion

Based on our study, we observed significant variations in the performances of different recovery strategies. Firstly, the recovery strategies based on link metrics and the product of the node centrality values of link endpoints did not achieve the highest performance. Secondly, the performance of those recovery strategies varied notably across different power systems, emphasizing the significance of network topology and generator placements in power grids. Thirdly, the two-step greedy recovery strategy consistently outperformed all other network metric-based recovery methods across all power systems, thresholds, and tolerance levels. The greedy recovery strategy, which was slightly less effective than the two-step greedy recovery strategy, also showed promising results. The findings confirm that implementing an effective recovery strategy can significantly improve the recoverability of power grids and that the two-step greedy strategy is particularly efficient, while relying solely on one network metric for recovery is less effective.

For future research directions, several avenues are worth exploring. Firstly, we can develop hierarchical recovery strategies, by classifying links into two groups—those connected to generators and those not connected to generators—and then prioritizing restoring links connected to generators. Secondly, we can investigate whether allowing cascading failures after link recovery or removal impacts the conclusions of this study. Thirdly, we can introduce additional constraints—such as generator costs—and incorporate cost minimization as an objective within the DC power flow model. Finally, given that distributed energy systems enhance power system resilience, it is promising to research how to establish and integrate such systems after a blackout, using the discussed strategies.

## Figures and Tables

**Figure 1 entropy-25-01455-f001:**
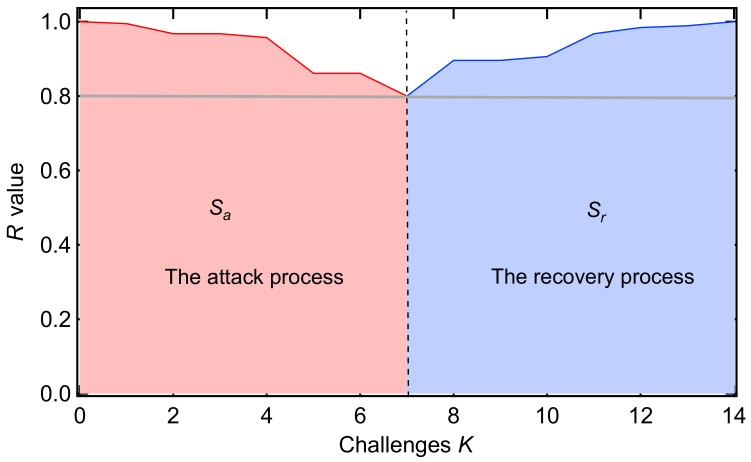
Illustration of the attack process and the recovery process in a power grid: one realization with a threshold equal to 0.8.

**Figure 2 entropy-25-01455-f002:**
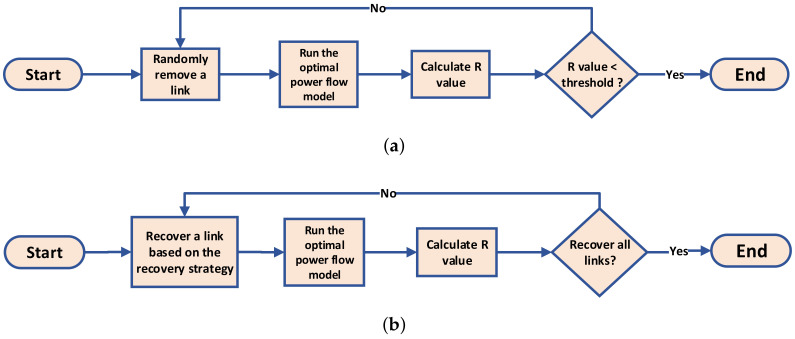
The flowcharts of the attack process and the recovery process: (**a**) the attack process; (**b**) the recovery process.

**Figure 3 entropy-25-01455-f003:**
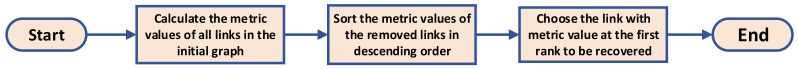
The flow chart of recovery strategies based on topological network metrics.

**Figure 4 entropy-25-01455-f004:**
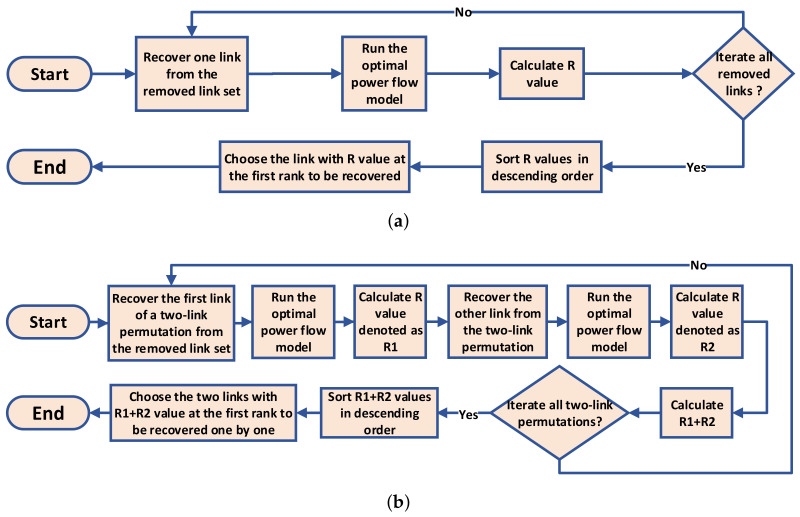
The flowcharts of the greedy and two-step greedy strategies: (**a**) greedy recovery strategy; (**b**) two-step greedy recovery strategy.

**Figure 5 entropy-25-01455-f005:**
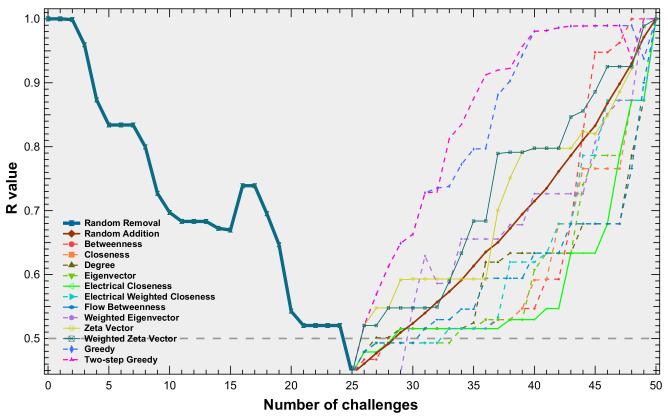
An example of *R* values during one random attack realization and of different recovery processes, with a threshold equal to 0.5 and a tolerance level equal to 2, for the IEEE 39 bus system. The results of the random recovery strategy are the average values of 100 realizations.

**Figure 6 entropy-25-01455-f006:**
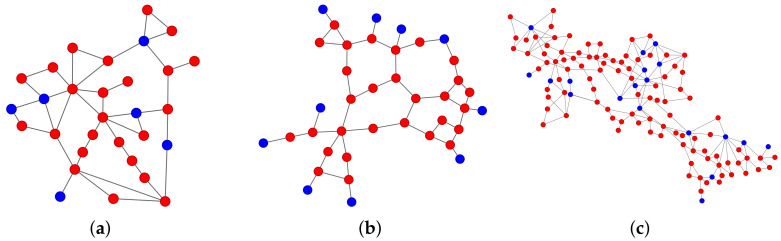
Three power grids. The red nodes represent loads and the blue nodes represent generators: (**a**) IEEE 30; (**b**) IEEE 39; (**c**) IEEE 118.

**Figure 7 entropy-25-01455-f007:**
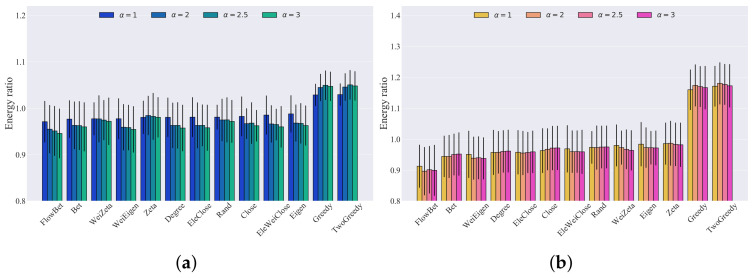
Bar graph of the different recovery strategies’ recoverability energy ratios, with thresholds equal to 0.8 and 0.5 for three bus systems with different tolerance levels. The number of realizations was 1000: (**a**) IEEE 30 threshold = 0.8; (**b**) IEEE 30 threshold = 0.5; (**c**) IEEE 39 threshold = 0.8; (**d**) IEEE 39 threshold = 0.5; (**e**) IEEE 118 threshold = 0.8; (**f**) IEEE 118 threshold = 0.5.

**Table 1 entropy-25-01455-t001:** Properties of three power grids.

Name	*N*	*L*	$d_{\mathrm{av}}$
IEEE 30	30	41	2.73
IEEE 39	39	46	2.36
IEEE 118	118	179	3.03

**Table 2 entropy-25-01455-t002:** The abbreviations of the strategies used in figures and tables in the paper.

Abbreviation	Full Name
TwoGreedy	Two-step greedy recovery strategy
Greedy	Greedy recovery strategy
Bet	Betweenness recovery strategy
FlowBet	Flow betweenness recovery strategy
EleWeiClose	Electrical weighted closeness recovery strategy
WeiEigen	Weighted eigenvector recovery strategy
Close	Closeness recovery strategy
EleClose	Electrical closeness recovery strategy
Rand	Random recovery strategy
Degree	Degree recovery strategy
Zeta	Zeta vector recovery strategy
WeiZeta	Weighted zeta vector recovery strategy
Degree	Degree recovery strategy
Eigen	Eigenvector recovery strategy

**Table 3 entropy-25-01455-t003:** The mean value and standard deviation values of the different recovery strategies for the IEEE 30 bus system, with the tolerance level α equal to 1, while we set the threshold values as 0.8 and 0.5.

Rank	Threshold = 0.8			Threshold = 0.5		
	**Strategy**	**Mean**	**Std**	**Strategy**	**Mean**	**Std**
**1**	TwoGreedy	1.0292	0.0241	TwoGreedy	1.1710	0.0656
**2**	Greedy	1.0285	0.0240	Greedy	1.1595	0.0654
**3**	Eigen	0.9877	0.0400	Zeta	0.9854	0.0679
**4**	EleWeiClose	0.9852	0.0420	Eigen	0.9837	0.0714
**5**	Close	0.9823	0.0427	WeiZeta	0.9796	0.0676
**6**	Rand	0.9808	0.0265	Rand	0.9733	0.0526
**7**	EleClose	0.9807	0.0429	EleWeiClose	0.9687	0.0762
**8**	Degree	0.9804	0.0423	Close	0.9629	0.0722
**9**	Zeta	0.9803	0.0358	EleClose	0.9576	0.0713
**10**	WeiEigen	0.9772	0.0440	Degree	0.9571	0.0727
**11**	WeiZeta	0.9772	0.0354	WeiEigen	0.9509	0.0761
**12**	Bet	0.9765	0.0404	Bet	0.9442	0.0667
**13**	FlowBet	0.9709	0.0447	FlowBet	0.9126	0.0693

**Table 4 entropy-25-01455-t004:** The mean value and standard deviation values of the different recovery strategies’ recoverability energy ratios for the IEEE 39 bus system, with the tolerance level α equal to 1, while we set the threshold values as 0.8 and 0.5.

Rank	Threshold = 0.8			Threshold = 0.5		
	**Strategy**	**Mean**	**Std**	**Strategy**	**Mean**	**Std**
**1**	TwoGreedy	1.0299	0.0221	TwoGreedy	1.1665	0.0605
**2**	Greedy	1.0292	0.0219	Greedy	1.1543	0.0585
**3**	Zeta	1.0065	0.0240	Zeta	1.0709	0.0571
**4**	WeiZeta	1.0000	0.0260	WeiZeta	1.0582	0.0594
**5**	Rand	0.9858	0.0217	Rand	0.9741	0.0474
**6**	WeiEigen	0.9800	0.0347	WeiEigen	0.9430	0.0618
**7**	Bet	0.9794	0.0333	Bet	0.9384	0.0673
**8**	EleWeiClose	0.9730	0.0347	EleWeiClose	0.8945	0.0625
**9**	Eigen	0.9703	0.0351	Eigen	0.8871	0.0653
**10**	Degree	0.9701	0.0363	Degree	0.8848	0.0706
**11**	FlowBet	0.9697	0.0331	FlowBet	0.8827	0.0618
**12**	Close	0.9675	0.0358	Close	0.8809	0.0629
**13**	EleClose	0.9646	0.0348	EleClose	0.8639	0.0621

**Table 5 entropy-25-01455-t005:** The mean value and standard deviation values of the different recovery strategies’ recoverability energy ratios for the IEEE 118 bus system, with the tolerance level α equal to 1, while we set the threshold values as 0.8 and 0.5.

Rank	Threshold = 0.8			Threshold = 0.5		
	**Strategy**	**Mean**	**Std**	**Strategy**	**Mean**	**Std**
**1**	TwoGreedy	1.0458	0.0270	TwoGreedy	1.2611	0.0715
**2**	Greedy	1.0441	0.0266	Greedy	1.2294	0.0717
**3**	Bet	0.9959	0.0409	Bet	1.0855	0.0683
**4**	FlowBet	0.9824	0.0685	WeiEigen	1.0225	0.0782
**5**	EleWeiClose	0.9759	0.0639	EleWeiClose	1.0206	0.0732
**6**	WeiEigen	0.9726	0.0648	Close	1.0051	0.0734
**7**	Zeta	0.9705	0.0388	FlowBet	0.9989	0.0988
**8**	Close	0.9703	0.0677	Zeta	0.9961	0.0735
**9**	Rand	0.9666	0.0400	Eigen	0.9909	0.0701
**10**	EleClose	0.9649	0.0799	Rand	0.9819	0.0545
**11**	WeiZeta	0.9611	0.0435	Degree	0.9646	0.0846
**12**	Degree	0.9599	0.0774	WeiZeta	0.9624	0.0715
**13**	Eigen	0.9523	0.0731	EleClose	0.9589	0.0855

## Data Availability

Not applicable.

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
