# Peer review of "Recovering Power Grids Using Strategies Based on Network Metrics and Greedy Algorithms"

_entropy, 2023, doi:10.3390/e25101455_

Round 1
Reviewer 1 Report
The recovering strategies in power grids is investigated in this study. Comments:
1. The literature review should be enriched by adding the relevant published papers. For example: WC Yeh, MF He, CL Huang, SY Tan, X Zhang, Y Huang, L Li. New genetic algorithm for economic dispatch of stand-alone three-modular microgrid in DongAo Island. Applied Energy 263, 114508, 2020.
2. The method used is not found in title
3. The abbreviations can only be mentioned at first appearance.
4. Please mention the main contributions item by item in the introduction.
5. Title of each table should be on the top of table.
Reviewer 2 Report
In the paper are presented efficient strategies for recovering individual links in power grids under link failures, that are focused on optimizing the DC power flow model to maximize the satisfaction of load demand after link removal or addition. However, minor revisions are required before it could be accepted for possible publication in the Entropy Journal.
Please consider the following remarks:
1. At first glance, the topic is of actuality and the state of the art is up-to-date with references form the last years. I recommend in Introduction section, Lines 48 – 74, to present more clearly the original contributions; maybe it would be better if the original contribution to be enumerated;
2. Sections 2 - 4 are well organized, presenting the mathematical modeling of power grids, with the attack and recovery process. Maybe, a flowchart can be included to describe all the steps from the efficient strategies proposed, not only for attack and recovery process.
3. The results are relevant, and the strategies are tested in three power systems, IEEE 30, IEEE 39, and IEEE 118. I appreciate the clear manner of presentation of the recovery strategies results.
4. The conclusions looks good.
Reviewer 3 Report
The study is well designed and conducted. The results are presented and analysed properly, and the conclusions address the limitations and extensions of the study correctly.
Some comments.
- The authors state in the abstract and conclusions that “the two-step greedy recovery strategy outperforms all other strategies”. Simultaneously, that “relying on a single metric for developing a recovery strategy is not enough.” Please clarify, since it seems that the “two-step greedy recovery” strategy should always be chosen.
- although fig. 5 suggests otherwise, the fact is that the values in tables 3, 4 and 5 are not too different, especially for the threshold 0.8. This would seem to imply that the strategies considered are practically equivalent, at least for the case of the threshold 0.8.
- It would be interesting to evaluate the applicability of these results to micro-grids in electricity generation, given the current trend towards distributed energy and all types of forms of shared self-consumption in small communities. Differentiating the large networks relevant to a country from the small ones, and determining whether the results are different or not would be important in the current context.
- The references are sufficient but somewhat outdated. I would encourage the authors to make an effort and update them.
- The English is ok but some expressions are a bit odd; please revise.
The English is ok but some expressions are a bit odd; please revise.
